

# A borehole trajectory inversion scheme to adjust the measurement geometry for 3D travel time tomography on glaciers

Sebastian Hellmann[1,2], Melchior Grab[1,2,3], Cedric Patzer[4], Andreas Bauder[1,5], and Hansruedi Maurer[2]

[1]Laboratory of Hydraulics, Hydrology and Glaciology (VAW), ETH Zurich, Zurich, Switzerland
[2]Institute of Geophysics, ETH Zurich, Zurich, Switzerland
[3]Terra Vermessungen AG, Othmarsingen, Switzerland
[4]Geological Survey of Finland (GTK), Espoo, Finland
[5]Swiss Federal Institute for Forest, Snow and Landscape Research (WSL), Birmensdorf, Switzerland

**Correspondence:** Sebastian Hellmann (sebastian.hellmann@erdw.ethz.ch)

**Abstract.** Cross-borehole seismic tomography is a powerful tool to investigate the subsurface with a very high spatial resolution. In a set of boreholes, comprehensive three-dimensional investigations at different depths can be obtained to analyse velocity anisotropy effects due to local changes within the medium. Especially in glaciological applications, the drilling of boreholes with hot water is cost-efficient and provides rapid access to the internal structure of the ice. In turn, movements of

the subsurface such as the continuous flow of ice masses cause deformations of the boreholes and exacerbate a precise determination of the source and receiver positions along the borehole trajectories. Here, we present a three-dimensional inversion scheme that considers the deviations of the boreholes as additional model parameters next to the common velocity inversion parameters. Instead of introducing individual parameters for each source and receiver position, we describe the borehole trajectory with two orthogonal polynomials and only invert for the polynomial coefficients. This significantly reduces the number

of additional model parameters and leads to much more stable inversion results. In addition, we also discuss whether the inversion of the borehole parameters can be separated from the velocity inversion, which would enhance the flexibility of our inversion scheme. In that case, updates of the borehole trajectories are only performed if this further reduces the overall error in the data sets. We apply this sequential inversion scheme on a synthetic data set and a field data set from a temperate Alpine glacier. With the sequential inversion, the number of artefacts in the velocity model decreases compared to a velocity inversion

without borehole adjustments and heterogeneities in the velocity model can be imaged similar to an inversion with correct borehole coordinates. Furthermore, we discuss the advantages and limitations of our approach in the context of an inherent seismic anisotropy of the medium and extend our algorithm to consider an elliptic velocity anisotropy. With this extended version of the algorithm, we analyse the interference between a seismic anisotropy in the medium and the borehole coordinate adjustment. Our analysis indicates that the borehole inversion interferes with seismic velocity anisotropy. The inversion can

compensate such a velocity anisotropy. Therefore, for such a borehole trajectory inversion, polynomials of degree three are a good compromise between a good representation of the true borehole trajectories and avoiding compensation for velocity anisotropy.



## 1 Introduction

Cross-borehole travel time tomography, based on seismic or ground penetrating radar (GPR) waves, is widely used to investi-
gate small-scale variations of the subsurface, especially when surface-based experiments lack from poor resolution. The main
advantage is the much higher ray coverage within the target area, allowing a more detailed analysis of small-scale variations
of geological heterogeneities. The first experiments with seismic sources have been described by Bois et al. (1972) and Menke
(1984) and successfully applied in a large number of studies in the last decades. Crosshole tomography has been used in min-
ing exploration (Wong, 2000), aquifer delineation and hydrology-related topics (Hubbard and Rubin, 2000; Daley et al., 2004;
Dietrich and Tronicke, 2009; Linder et al., 2010; von Ketelhodt et al., 2018), for characterising a host rock and its small-scale
structures such as fault planes (Maurer and Green, 1997; Rumpf and Tronicke, 2014; Schmelzbach et al., 2016; Doetsch et al.,
2020; Shakas et al., 2020), for investigating pore pressure variations in aquifers and caprocks (Daley et al., 2008; Marchesini
et al., 2017; Grab et al., 2022) and voids in karst regions (Duan et al., 2017; Kulich and Bleibinhaus, 2020; Peng et al., 2021).
Furthermore, Gusmeroli et al. (2010) and Axtell et al. (2016) used GPR-based travel time tomography to estimate the water
content in a polythermal glacier.

In contrast to surface-based experiments, the sources and receivers are usually not directly accessible. In general, the borehole
geometry is assumed to be well-known, which allows a correct calculation of distances between source and receivers. For this
purpose, inclinometer and caliper measurements are required to describe the actual borehole geometry, and centralisers have
been used to precisely position the tools in the centre of the boreholes. Nevertheless, there are some applications, where such
estimates are not feasible. For example, for glaciological applications as described in Axtell et al. (2016), boreholes are usually
drilled by using hot-water (Iken, 1988) or steam drills (Heucke, 1999). These are a highly efficient and cheap methods for
glaciology-related investigations, but they come at the cost that the boreholes have a variable diameter. The upper part of the
hole is exposed to hot water for a much longer time than the deepest parts of the boreholes leading to a rather conical shape of
the boreholes. Furthermore, outbreaks due to impurities, meltwater intrusions and air content that change the thermal capacity,
and a variable drilling speed further complicate the borehole shape. In those boreholes, centralisers can usually not be used and
the actual position of the instruments may vary around the assumed position (Schwerzmann et al., 2006; Axtell et al., 2016).
Furthermore, glaciers constantly move during the time of measurements and especially for comprehensive 3D experiments,
which may last days or weeks, this leads to larger deformations of the boreholes over time and the assumed distances are no
longer correct. Although repeated inclinometer measurements may reduce the errors, the required precision may still not be
reached.

Dyer and Worthington (1988) and later Maurer (1996) have already shown the significant effect of wrong source and receiver
coordinates on travel time inversions. Artefacts have been introduced in the resulting velocities leading to a misinterpretation
of the measurements. They proposed adjusted inversion algorithms that consider the coordinates as additional model parame-
ters and also inverts for the coordinates in a 2D inversion. More recently, Kim and Pyun (2020) developed an approach using
full-waveform inversion and a grid-search algorithm to update the 2D receiver coordinates after each step of velocity inversion
for a data set from vertical seismic profiling (VSP) experiments. Coordinates and velocity are inverted in two separate steps




allowing a more focused adjustment of different physical parameters.

In general, the majority of tomography measurements only consider 2D applications. This requires a calculation of a 2D tomographic plane that contains the boreholes. Irving et al. (2007) developed a coordinate rotation algorithm to minimise sum of

out-of-plane deviations for sources and receivers. Nevertheless, 2D tomography results do not consider off-plane effects and even when the results of several 2D planes are combined in the end, the individual inversions do not account for the whole 3D information. Therefore, several 3D inversion schemes have been developed in the last years (Washbourne et al., 2002; Cheng et al., 2016), showing the advantages of a 3D inversion over a combination of individual 2D inversions.

However, for 3D inversions, borehole trace corrections are more complex due to the additional degrees of freedom. Bergman

et al. (2004) proposed a method for 3D tomography experiments that implements a static correction term for each receiver to absorb short-wavelength variations in layered subsurface, but other approaches that are successful for 2D inversions (e.g. Maurer, 1996) are difficult to transfer to 3D as the inversion scheme is severely under-determined. This may lead, for example, to oscillations of the coordinates at larger depths.

This issue becomes even more complicated, when considering a subsurface with a certain velocity anisotropy. In glaciers, the

ice crystals interact with the flow of the ice mass. The strain conditions in the glacier force the ice crystals, that is, the $c$ axes as the symmetry axis of the hexagonal ice grains, to align in certain crystal orientation fabric (COF) pattern in accordance with the given stress conditions (e.g. Duval et al., 1983; Alley, 1988). In a compressional regime in polar ice, single maxima have been observed in a variety of ice cores and geophysical experiments (e.g. Azuma and Higashi, 1985). The COF-derived seismic anisotropy effect was initially investigated by Bentley (1972). Diez and Eisen (2015) analysed the effect on seismic velocities

for different COF pattern and found an anisotropy of up to 6 % for p-waves. For temperate glaciers, older and more recent studies have observed multi-maxima for compressional regimes (Kamb, 1959; Rigsby, 1960; Hellmann et al., 2021b; Monz et al., 2021). Hellmann et al. (2021a) have calculated an azimuthally dependent anisotropy of 2.3 % as a result of such a multi-maximum. Media with such a degree of anisotropy are considered as weakly anisotropic. Especially (Thomsen, 1986) analysed the effect of weak anisotropy on seismic wave propagation and provided linearised equations for seismic velocities by intro-

ducing a set of five parameters describing the strength and direction of anisotropy. Later, other studies developed approaches to further describe the anisotropy in specific media with certain anisotropy pattern or fracture orientations and provided ways to implement such an anisotropy in existing algorithms (e.g. Faria and Stoffa, 1994; Zhou and Greenhalgh, 2005; Martins, 2006; Chen et al., 2017; Meléndez et al., 2019; Pan et al., 2021). In particular, Zhou et al. (2008) demonstrated the effect of anisotropy on crosshole traveltime tomography results for tilted transversely isotropic media. The outcome of all these studies

is, that velocity anisotropy has to be considered to have an influence on the velocities obtained from a travel time tomography. In this study, we propose an approach for a travel time tomography that is explicitly designed for the glaciological experiments and apply this to a multi-crosshole seismic data set acquired on Rhonegletscher as a case study. The continuous but not necessarily linear movement and ice melt of the glacier during data acquisition caused a deviation of the holes leading to uncertainties about the particular source and receiver coordinates. Inclinometer measurements at the beginning and end of

the campaign provide some constraints considered in the inversion. Instead of inverting for the individual source and receiver positions, we assume that the borehole trajectories can be described with two perpendicular 2D polynomials $x(z)$ and $y(z)$.



The degree $n$ of the polynomials is used to control the degree of variations along the trajectories. The additional inversion parameters are these polynomial coefficients. Similar to the approach of Kim and Pyun (2020), we then employ a two-step inversion algorithm that inverts for the velocity and the coefficients of the polynomials in alternation. We further investigate

and discuss the interdependence between borehole adjustments and the inherent weak velocity anisotropy of the medium.

## 2   Methodology

The inversion scheme that we apply in this study consists of two parts. The first step is a three-dimensional velocity inversion, based on common ray-tracing performed on a finite difference grid. Each cell $j$ (i.e., rectangular prism) has a defined velocity, expressed by its reciprocal value of slowness $s_j$. For a given source-receiver combination $i$, the travel time through a cell can

be calculated from the length $l_{i,j}$ of the ray through that cell and its associated slowness value $s_j$. Summing over all $N$ cells between this source-receiver pair results in the travel time $t_i$, defined as

$$t_i = \sum_{j=1}^{N} l_{ij} s_j. \tag{1}$$

With the measured travel times from cross-borehole experiments $\boldsymbol{t}^{\text{obs}}$ and an initial start model $\boldsymbol{m}^{\text{ini}}$ for the velocity distribution and the resulting calculated travel times $\boldsymbol{t}^{\text{cal}}$, the velocities of the cells that are covered by ray paths, which are determined by a

ray tracing algorithm, can be updated iteratively,

$$\Delta \boldsymbol{d} = \boldsymbol{t}^{\text{obs}} - \boldsymbol{t}^{\text{cal}} \tag{2}$$

$$\Delta \boldsymbol{m} = (\mathbf{G}^{\text{T}} \mathbf{G} + \mu \mathbf{W}_M)^{-1} \mathbf{G}^{\text{T}} \Delta \boldsymbol{d} \tag{3}$$

$$\boldsymbol{m}^{\text{new}} = \boldsymbol{m}^{\text{ini}} + \Delta \boldsymbol{m}, \tag{4}$$

where the matrix $\mathbf{G}$ contains the partial derivatives (sensitivities) $G_{ij} = \frac{\partial d_i}{\partial m_j}$ for each velocity grid cell. The matrix $\mathbf{W}_M$ and

$\mu$ summarise the regularisation (i.e. damping and smoothing) that is required for a stable inversion. Further details about the regularisation are provided, for example, in Maurer et al. (1998).

The second part of our inversion scheme contains a coordinate inversion that accounts for uncertainties about the borehole trajectories. Although inclinometer data are usually available, specific conditions such as a variable borehole diameter or a moving subsurface incorporate uncertainties that significantly affect the velocity inversion and may lead to artefacts complicating an

interpretation. An inversion scheme for the individual coordinates $x, y, z$ for each source and receiver would result in a severely under-determined problem. This approach with $3(n_S + n_R)$ unknowns can be replaced by a more robust one, when assuming that the sources and receivers in a borehole are aligned along the continuous borehole trajectory. Then, displacements of the sources and receivers can be described by deformations of the trajectory. As a simple mathematical description, we assume that each three-dimensional borehole trajectory can be described by two orthogonal two-dimensional polynomials $x = p(z)$

and $y = q(z)$. The maximum degree $n_p$ of each polynomial is selected by the user according to a priori information or an educated guess. The inversion problem assumes a known mean velocity between each source and receiver, e.g. derived from





the estimated velocity from the previous step in the velocity inversion. Then, a second inversion scheme like the one defined in (3) can be set up. The Jacobian matrix $\mathbf{J}$ contains the first derivatives of the data against the new model parameters, i.e. the polynomial coefficients $A_j$, $j = 1, ..., n_p$,

$$J_{ij} = \frac{\partial d_i}{\partial A_j} = \frac{\partial d_i}{\partial r_k} \frac{\partial r_k}{\partial A_j}, \tag{5}$$

with $k = 1, 2$ for the two horizontal and linearly independent directions $x = r_1$ and $y = r_2$. For instance, for the polynomial $x = p(z)$ of the source hole

$$x = p(z) = \sum_{j=1}^{n_p} A_j (z - z_0)^j \tag{6}$$

the first derivatives ($j = 1, ..., n_p$) for the $i^{\text{th}}$ combination of the source $(x_S, y_S, z_S)$ and receiver positions $(x_R, y_R, z_R)$ are

$$J_{ij} = -\frac{1}{v} \frac{x_R - x_S}{\sqrt{(x_R - x_S)^2 + (y_R - y_S)^2 + (z_R - z_S)^2}} (z_S - z_0)^j, \tag{7}$$

where $v$ is the mean velocity between the source and the receiver. $(x_0, y_0, z_0)$ is the collar point of the borehole and assumed to be known, e.g. from differential global navigation satellite system (GNSS) measurements, so that the constant term $A_0 \equiv 0$. In addition, the relative position of the sources and receivers along the boreholes is assumed to be known, for example, by measuring the length of the cables of the instruments in the borehole below the collar point. Similarly, this framework can be applied to the polynomial $y = q(z)$. For the receiver borehole, $\mathbf{x}_R$ and $\mathbf{x}_S$ need to be exchanged. The regularisation term $\mu \mathbf{W}_M$ in (3) is exchanged with a Tikhonov regularisation, i.e. $\mathbf{W}_J = \mu \mathbf{I}$, with a damping factor $\mu$ that can individually be adjusted for each borehole.

There are two options for implementing the two parts of the inversion. The first one is a sequential inversion. During each iteration of the inversion, the updates of velocity and borehole trajectories are computed independently. The second option is an extended system of equations that considers both parts in one large Jacobian matrix. We have tested both options and could not find significant differences between both methods for synthetic data sets. However, the sequential inversion seems to be numerically more stable. Furthermore, the most recent update of the velocities is already considered in the current step of iteration and additional constraints determining whether a borehole trajectory inversion should be performed are easier to evaluate. Therefore, we calculated the results in the next sections with the sequential inversion.

## 3 Experimental setup for the field data acquisition on Rhonegletscher

Initially we encountered the issue of deviating boreholes, when acquiring cross-borehole seismic data for anisotropy-related investigations on Rhonegletscher (Rhone glacier), a temperate glacier (ice temperature $T \approx -0.5°\text{C}$) in the Swiss Alps. The glacier still covers an area of $15 \, \text{km}^2$ and is flowing in southern direction at its current terminus (GLAMOS, 2020). For anisotropy investigations, we drilled a set of 13 boreholes (Fig. 1a) about $500 \, \text{m}$ north of the terminus with a hot-water drilling system (e.g. Iken, 1988) in summer 2018. The boreholes were arranged in a ring with a diameter of $40 \, \text{m}$ to obtain cross-borehole seismic measurements under different azimuths (i.e. 0, 30, 60, and 90° relative to the ice flow direction). At the





borehole location, the glacier was about $100\,\mathrm{m}$ thick and the ice flew with a surface flow speed of $16\,\mathrm{m\,a^{-1}}$ in southeastern direction (Hellmann et al., 2021b). This ice flow velocity decreases with depth which leads to a continuous deformation of the boreholes over time. At the surface, the ice moved by $1.2\,\mathrm{m}$ within the three weeks of data acquisition. The position of the

borehole collars and additional geophones along the surface were measured using a high precision Leica GNSS device in RTK mode. In addition, we used an inclinometer probe from Geotomography GmbH for an estimate of the borehole trajectories after drilling and observed the ongoing deformation with two repeated measurements. An example from borehole BH01 is shown in Fig. 1b. The continuous deformation could not fully be surveyed by just three inclinometer measurements and, thus, leads to unknown borehole trajectories for intermediate time steps of data acquisition. Nevertheless, the inclinometer measurements

could serve as initial estimates for a borehole trajectory inversion.

The drilling of the boreholes was stopped about $15\text{-}20\,\mathrm{m}$ above the glacier bed to avoid hydrological coupling of the borehole with the subglacial drainage system resulting in a total borehole length of $80\text{-}90\,\mathrm{m}$. For the experimental set up, the boreholes needed to be water-filled. Despite this precaution, borehole BH09 and a second hole just a few metres next to it drained a few minutes after drilling. Similarly, borehole BH11 drained after the second out of three days of measurements in this borehole.

Since the seismic sparker source and hydrophone receivers can only be deployed in water filled boreholes, this drainage led to an incomplete data set.

A $5\,\mathrm{kV}$ sparker source from Geotomography GmbH with a dominant frequency of $2\,\mathrm{kHz}$ was employed in the boreholes for generating the acoustic signal. Hydrophones were installed in the borehole on the opposite side of the ring, as well as in a second borehole either perpendicular or parallel to the glacier flow (Fig. 1a). Geophones at the surface of the glacier complemented the

measurement setup. This connection of each borehole to two other holes is a prerequisite for the borehole trajectory inversion. Therefore, the exclusively two-dimensional data sets acquired between BH06 and BH12 as well as BH03 and BH00 could not be considered as explained in more detail below.

For a sufficiently dense data coverage, we selected a shot interval of $1\,\mathrm{m}$ in the source holes as well as a distance of $1\,\mathrm{m}$ between the receivers along the surface and within the receiver boreholes. At least three shots were recorded and stacked during the

processing to enhance the signal to noise ratio. Since the length of the hydrophone chain was limited to $23\,\mathrm{m}$ (24 channels), the experiment had to be repeated four times to cover the entire length of the receiver borehole. For the data processing, we applied a standard procedure consisting of a median and bandpass filtering ($0.3 - -15\,\mathrm{kHz}$) to enhance the signal quality, a stacking of repeated shots, and a picking of the p-wave first arrivals with a cross-correlation algorithm. To pick the onset of the p-wave, we defined a window around the expected first break arrival time and analysed the coherency between the traces

within the individual receiver gathers. Finally, we performed different combinations of 3D velocity and borehole inversions.

## 4 Application to a synthetic data set

To demonstrate the effect of our borehole correction, we first applied the approach to a synthetic cross-hole seismic data set. For this purpose, we defined a set of nine boreholes with a length of $80\,\mathrm{m}$. The position of the borehole collar points were taken from the actual field measurements described in Sect. 3. For the deviations of the boreholes, we fitted a $4^{\mathrm{th}}$ order polynomial





through the data points of the inclinometer measurements and exaggerated this deviation by a factor of 8 to investigate the reliability of our code for cases with even more prominent deviations, e.g. for glaciers with even higher ice flow velocities than those observed on Rhonegletscher. We placed 80 sources and 80 hydrophones in each borehole and added a total of 828 receivers on an inclined plane that represents the surface. The geophone positions also refer to GNSS field measurements and were placed $12-20$ m below the reference point of the model. The measurement geometry (i.e. the selection of source-receiver

profiles) for the synthetic calculations was also based on real field measurements for developing a tool that is suitable for our actual field measurements with a limited number of source-receiver combinations. Therefore, each source hole was connected with only two receiver holes similar to the profiles shown in Fig. 1a.

For the investigations, we defined a heterogeneous velocity model (shown in Fig. 2) that consists of the following parts: The background velocity of the true model was set to $3800 \, \mathrm{m \, s^{-1}}$. This value has been measured in ice core samples from Rhone-

gletscher (Hellmann et al., 2021a) and the cross-borehole field experiments were carried out at the same location. Therefore, we also used this value as background velocity for our synthetic example. Furthermore, we added two north-south oriented fault zones (e.g. water-saturated intrusions in the ice) with a slightly lower velocity of $3740 \, \mathrm{m \, s^{-1}}$ close to but below the geophones ($z = 20-40$ m of the model). In addition, two meandering, inclined structures (representing englacial channels) with a velocity of $3680 \, \mathrm{m \, s^{-1}}$ were included at $z = 40-60$ m and $z = 70-86$ m. Synthetic data were computed without additional

random noise, and they were subsequently inverted using a homogeneous start model with a velocity of $3800 \, \mathrm{m \, s^{-1}}$. Damping and smoothing parameter for the regularisation were selected by trial and error (smoothing factor of 0.4, damping factor of 0.1 for the entire model). The results of the velocity inversion with correct source and receiver positions are shown in Fig. 3a. The comparison with the true model in terms of velocity differences between inverted and true model is shown in Fig. 3b. The heterogeneities of the true model could be resolved quite well within the ring of boreholes, that is, the area that is covered

by the ray paths of the measurements. This model is regarded as reference model in the following discussion, although the thickness of the two channels in the lower part of the model and the velocity of the channel between 40 and 60 m are still slightly overestimated. However, these artefacts are most likely a result of the smoothing constraints in the velocity inversion and therefore beyond the scope of our study.

In a next step, the information about the true borehole inclination was ignored and we started with straight vertical boreholes.

We ran the inversion without and with borehole trajectory inversion and stopped the inversions after seven iteration steps. Table 1 provides the root mean squared (RMS) errors for both inversion schemes. The values provide evidence that these schemes both converge, but the RMS values of the sequential inversion are significantly smaller, thus providing a better fit between calculated and measured travel times. Figure 4a shows the inverted velocity model for a pure velocity inversion. The differences

**Table 1.** RMS errors (RMSE) for the velocity and the combined inversion.

| iteration | 1 | 2 | 3 | 4 | 5 | 6 | 7 |
|---|---|---|---|---|---|---|---|
| RMSE (only velocity inv.) [ms] | 0.0785 | 0.0507 | 0.0386 | 0.0378 | 0.0374 | 0.0372 | 0.0372 |
| RMSE (combined inv.) [ms] | 0.0785 | 0.0304 | 0.0216 | 0.0200 | 0.0192 | 0.0187 | 0.0186 |





to the reference model (Fig. 3) are shown in Fig. 4b. The velocity of the upper and especially the lower channel between 70

and 86 m could not be recovered incorporating clear artefacts in the inversion results (blue dashed ellipses A+B in Fig. 4b). In addition, several artefacts appear at the bottom of the model around the boreholes (see the blue solid ellipses C+D in Fig. 4b showing very prominent examples). This is the region that is covered by only a few measurements and therefore highly under-determined. Furthermore, the discrepancy between the true inclined boreholes and the assumed straight borehole is largest at the bottom of the boreholes. As a result, geometrical errors are smeared into the velocity distribution in the least resolved parts

of the model. We also refer to the supplementary video material for an enhanced overview about the velocity distribution and artefacts in the ice volume.

The calculations for the inversion were repeated with an additional borehole trajectory inversion. Figure 5a shows the results of this inversion and Fig. 5b shows the differences with respect to the reference model. Here, the upper and especially the lower channel (ellipses A+B) are correctly resolved with a velocity misfit of $< 20\,\mathrm{m\,s^{-1}}$ inside the channel. In addition, the prominent

artefacts in the vicinity of the boreholes and artefacts at the bottom of the model (ellipses C+D) are significantly reduced. Thus, including a trajectory inversion yields results comparable with those in Fig. 3 obtained by using the true borehole geometry.

The borehole trajectories, obtained from the trajectory inversion, are shown in Fig. 6 and compared with initial (straight) and true trajectories. For some boreholes, the inverted borehole trajectory closely resembles the true borehole path. For other boreholes, e.g. borehole BH02 or BH07, the inversion algorithm did not fully converge towards the true solution. Instead, some

minor artefacts are still visible in the velocity profile in the vicinity of these boreholes (for an enhanced three-dimensional view we refer to our video supplement). This is a trade-off that cannot fully be avoided by such a velocity-borehole inversion as both solutions provide similar residual errors. If a priori information about the magnitude of deformation or an initial guess for the main direction of deformation are available, this information could be used to avoid a start with straight vertical boreholes and provide further constrains for the individual boreholes. However, boreholes that are only part of a single two-dimensional

profile cannot be fitted by our algorithm at all, as the degree of freedom is then higher than the geometric information obtained from such a two-dimensional seismic profile (e.g. dashed profile BH06-BH12 in Fig. 1a). In this case, we observed strong oscillations or large deviations of the trajectories perpendicular to the measurement plane.

The issue of resolving the individual model parameters can be further investigated with the resolution matrix $\mathbf{R}$ (e.g. Menke, 1984),

$$m^{\mathrm{est}} = \mathbf{R}m^{\mathrm{true}} \tag{8}$$

that connects the true model parameter $m^{\mathrm{true}}$ with the estimated parameters $m^{\mathrm{est}}$. The resolution matrix for our velocity-borehole inversion scheme is defined as

$$\mathbf{R} = \left( \begin{pmatrix} \mathbf{G} & \mathbf{J} \\ \mathbf{W}_M & 0 \\ 0 & \mathbf{W}_J \end{pmatrix}^T \begin{pmatrix} \mathbf{G} & \mathbf{J} \\ \mathbf{W}_M & 0 \\ 0 & \mathbf{W}_J \end{pmatrix} \right)^{-1} \left( \begin{pmatrix} \mathbf{G} & \mathbf{J} \end{pmatrix}^T \begin{pmatrix} \mathbf{G} & \mathbf{J} \end{pmatrix} \right), \tag{9}$$

where $\mathbf{G}$ is the sensitivity matrix (Jacobian matrix) for the velocity inversion, $\mathbf{J}$ is the Jacobian matrix for the borehole inversion

as defined in equation 5. The regularisation matrices (combining damping and smoothing) are given by $\mathbf{W}_M$ for the velocity



inversion and by $\mathbf{W}_J$ for the borehole trajectory inversion, respectively. The resolution matrix for the synthetic example in Fig. 5b is shown in Fig. 7. Only values larger than 0.05 are plotted as black dots. The matrix has a tridiagonal structure with the highest values along its main diagonal. Therefore, the estimated model parameter is mainly dependent from its true value. However, for the velocity inversion parameters, this dependency is rather weak (all values are < 0.2). In addition, elements
along two minor diagonals are also influencing the estimated parameters. These are the neighbouring model parameters along the ray path that also affect the velocity of the current cell.

For the borehole inversion parameters, the relationship between estimated and true parameters is much stronger, especially for the polynomial coefficients of degree one and two (Fig. 7b). Here, values > 0.99 show that these coefficients are fully resolved in the inversion. However, the higher-order polynomial coefficients (3rd+4th degree) are also weakly resolved for the majority
of the boreholes (small values for these coefficients in Fig. 7b). This indicates that the exact values for the coefficients cannot be determined independently and several solutions lead to similar results. The borehole coefficients of a particular borehole are also dependent from the coefficients of the other boreholes and its higher order coefficients are often accompanied by larger off-diagonal elements that indicate this dependency. Here, the most prominent off-diagonal elements that are visible in Fig. 7b are part of the other component of the borehole (i.e. the higher-order coefficients of the polynomial $q(z)$ interfere with
the coefficients of $p(z)$) indicating that the two polynomials are not fully linearly independent. This is not surprising, when considering the geometry of the experimental setup (Fig. 1a) that is based on field measurements with a limited number of source-receiver combinations. The data set for the three-dimensional inversion consists of a set of two-dimensional profiles covering the volume within the ring of boreholes. In this setup, each borehole is used as source and receiver hole together with two other boreholes. Therefore, a minimisation of the error can be achieved by adjusting at least one of the polynomials of
these boreholes relevant for the current set of measurements. This leads to a reduced azimuthal illumination of the borehole trajectories and several equally good solutions for the borehole trajectories are possible and reduces the number of fully independent model parameters.

The resolution matrix is also a useful tool to investigate if the model parameters of the velocity inversion also affect the borehole trajectory inversion and vice-versa. If there is a dependency between these parameters we expect to see off-diagonal
elements in the resolution matrix. As shown in Fig. 7a, off-diagonal elements appear in the lower-left corner of the matrix, but not in the upper-right part. This implies that the estimated borehole polynomials depend on the defined velocity parameters between the boreholes. This is not surprising since the velocity is included in the calculation of the sensitivities for the borehole inversion and has already been investigated by Maurer (1996). In contrast, there is no dependence of the velocity parameters on the selected coefficients of the borehole trajectories. Therefore, the velocity inversion can be obtained independently from
the borehole parameters and a separate borehole trajectory inversion is acceptable, if the most recent results for the velocities are considered in the iteration step of the borehole inversion.




## 5 Application on field data

For the field data set, we determined the travel times of the recorded p-waves with a cross-correlation algorithm between repetitive measurements (in general 3 repetitions) within a 200 ms window around the estimated arrival time. Afterwards, these

absolute values of the picked travel times were compared. If they differed by less than 3 samples (or 60 µs) the median value of the arrival times was selected for the source-receiver pair. About 7.2% (25860 out of 359369) of all source-receiver pairs had to be excluded from the analysis due to larger discrepancies. In a next step, we obtained a velocity inversion followed by a combined velocity and borehole inversion. The results for the velocity inversion are shown in Fig. 8a. Similar to our synthetic data, we observe very locally high velocities (yellow ellipses). We interpret them as artefacts since in previous experiments

on ice core samples (Hellmann et al., 2021a) we observed a mean seismic velocity of $3820\,\mathrm{m\,s^{-1}}$ in pure ice at -5°C with less than 2% air bubbles. We are not expecting to reach such high values for in situ measurements due to the high amount of melt water that is present in the entire glacier in summer. In addition, we also observe velocity artefacts at the lower end of several boreholes similar to those in our synthetic example. Therefore, we applied our borehole trajectory inversion scheme in addition to the velocity inversion. The results are shown in Fig. 8b. Both inversion models show a low-velocity zone close to the

surface, which we interpret as a zone of weathered, water-saturated ice. The meltwater, generated at the surface of the glacier due to the positive energy flux into the ice, leads to a decrease of the seismic velocities. Between 15 and 70 m, we observe a rather homogeneous zone with a velocity of $v = 3750 \pm 30\,\mathrm{m\,s^{-1}}$. These values lie in the range of values commonly observed in glacier ice (Deichmann et al., 2000; Walter et al., 2009; Church et al., 2019). Similarly, in the lower part of the glacier an englacial drainage system, observed in the vicinity of our investigation area by Church et al. (2020, 2021), also results in larger

amounts of water. Therefore, the velocity decreases again.

The RMS values of both inversion schemes are similar as shown in Table 2. As a consequence, the velocity structure resulting form the two inversion schemes does not significantly differ. Nevertheless, velocity artefacts visible in the velocity inversion vanish in the combined inversion depending on the selected damping factor. This damping factor also affects how strongly

**Table 2.** RMS errors (RMSE) for the velocity and the combined inversion applied to the field data.

| iteration | 1 | 2 | 3 | 4 | 5 | 6 | 7 |
|---|---|---|---|---|---|---|---|
| RMSE (only velocity inv.) [ms] | 0.192379 | 0.130918 | 0.120189 | 0.118738 | 0.118036 | 0.117659 | 0.117347 |
| RMSE (combined inv.) [ms] | 0.192379 | 0.126254 | 0.118537 | 0.116894 | 0.116199 | 0.115779 | 0.115507 |

the adjusted boreholes deviate from the initial guess. Too high values (damping factor $\geq 10000$) do not significantly change

the borehole trajectories and the artefacts are still in place. A very low damping factor ($< 1000$) leads to unrealistically large adjustments with up to several metres at the bottom of the borehole. Therefore, we selected a damping factor, that provided the best compromise between adjustment of boreholes and removing the velocity artefacts. Furthermore, we considered different start values for the trajectories to test the robustness of our algorithm. Keeping all other regularisation factors identically, we started with perfectly vertical boreholes and already inclined boreholes. The latter are interpolated trajectories (by means of

time) derived from the repeated inclinometer measurements. Figure 8b shows the results for an inversion with initially vertical



boreholes. The resulting adjustments of the trajectories for both start models are compared in Fig. 9. The trajectories are consistent providing evidence that the algorithm provides robust results. This is especially useful if no a priori (i.e. inclinometer) measurements of the trajectories would be available. In addition, the results of the majority of the boreholes are consistent with regard to the time difference between drilling and day of measurement. The boreholes BH02, 04, 08, and 10 were occupied by

the instruments about 20-23 days after drilling and show larger deviations compared to the holes BH01 and 05, which were occupied 11-14 days after drilling. The ongoing ice flow causes a slightly enhanced deviation at a later stage.

Nevertheless, the boreholes BH07 and BH11 show large deviations. Their maximum deviation at the bottom is 0.6 and 1 m, respectively, although the measurements were obtained just 11-14 days after drilling of the holes (together with BH01 and 05). Glacier flow rates of $0.06\,\mathrm{m\,d^{-1}}$ on average measured at the glacier surface during the summer period imply that this

deviation could not be caused by the ice flow. Furthermore, the inclinometer measurements do not provide any hint for such strong deviations. Therefore, we expect that measurement errors such as picking errors are compensated through over-adjusted borehole trajectories during the inversion. Further evidence for this assumption is given by the fact that BH11 was only used as a source hole and only partially as receiver hole (along a profile to BH07 but not to BH05), since it drained just after the second out of three days of measurements. Therefore, the number of data points in this borehole is much lower and, thus, the

capability of our algorithm decreases due to poor constraints by the data. Since BH07 is directly connected to BH11 with two reciprocal profiles, errors in the trajectory of BH11 may also affect BH07 and would explain the still quite large deviations.

In summary, our inversion algorithm provides the option to adjust borehole trajectories such that artefacts in the velocity inversion can be removed. For a good three-dimensional adjustment, a reasonably high number of data points along these trajectories for different directions is required. For our specific experimental setup on a continuously moving glacier, this

inversion scheme provides improved results.

## 6 Discussion

The combined inversion scheme provide the advantage of a subsequent correction for the borehole coordinates if no or limited information about the positions of sources and receivers is available. However, there is a risk that the coordinate adjustment will suppress the appearance of real velocity anomalies in the tomogram. We avoid this by decoupling the two parts of the

inverse algorithm. Furthermore an additional check, whether the new coordinates reduce the RMS error of the entire data set, was implemented. This led to a more cautious adjustment. For our field data set, only 2-3 coordinate adjustments in the first iterations are required. Afterwards, the inversion continued with a traditional velocity inversion and explained the rather similar RMS values observed in Table 2. This is the major advantage of the sequential inversion scheme with two independent inversion branches compared to an "all in one" inversion. Nevertheless, the borehole trajectory inversion still relies on a priori

knowledge such as inclinometer estimates and must be tuned accordingly to find the best solution for this equifinality problem.





## 6.1 Comparison with previous studies

The idea for a sequential inversion scheme is mainly driven by the findings described in Maurer (1996) and following projects in the research group. Two-dimensional analyses are usually easier to evaluate as the degree of freedom is lower that in a three-dimensional setting. However, as Irving et al. (2007) already described in detail, off-plane effects may lead to misinterpretations

of the resulting data. Especially when analysing an azimuthally dependent variation in the seismic velocities (for example, due to changes in the ice crystal orientation relative to the glacier flow), those off-plane effects have to be avoided. Cheng et al. (2016) described a similar three-dimensional issue, when trying to detect boulders in the subsurface. With their numerical modellings, they presented the advantages of the three-dimensional algorithms with the resulting higher capability. Due to these advantages of three-dimensional approaches, we considered them here as well.

Next to the issues in the velocity model, we also have to consider the ice flow and thus a changing geometry setup, when obtaining the inversion. Once again, some successful concepts were described in earlier studies. As an example, Bergman et al. (2004) presented their results for a layered subsurface. They applied some static corrections to account for the layers and changes in between. Although glaciers may also have some ice layers, such as air bubble free and air bubble rich layers (Hubbard et al., 2008), the current study is mainly driven by issues due to the glacier flow itself and the consecutive deformation

of the initially rather straight boreholes in the glacier. Therefore, we could not consider such an approach for our data analysis. In one of the most recent publications, Fernandes and Mosegaard (2022) provided a promising statistical concept in which they determine the probability of the borehole positions along a given trajectory by perturbating the model parameters. This might be another promising approach that we did not consider. However, there might be some issues due to either a very large number of sources and receivers and a poorly constraint initial borehole trajectory that even further deforms over time. As an

alternative, we considered the methodology described by Maurer and Green (1997). They analysed the effect of coordinate mislocations on crosshole tomography results in 2D. Their findings were similar to our observations after data acquisition and processing. We therefore considered their concept and transferred it into 3D.

## 6.2 Interference of the borehole inversion scheme with velocity anisotropy

An adjustment of the borehole coordinates provides the opportunity to account for deformations of the boreholes due to a

movement of the subsurface. This adjustment bases on the current velocity model as it considers the mean velocity between source and receiver. However, other physical quantities such as a seismic anisotropy may introduce apparent errors in this seismic velocity values. In glaciers, macrostructural features, such as the crevasses or englacial channels, used in our synthetic data set, may cause a velocity anisotropy in the seismic data. Besides this macrostructure, there is the crystal orientation fabric that can also introduce such an anisotropy. This anisotropy appears since the seismic velocity in a single ice crystal depends

on the propagation direction of the seismic wave relative to the $c$ axis of the crystal. When the ice crystals in a polycrystalline glacier are oriented in a preferred direction relative to given stress conditions, a seismic wave that travels parallel to this alignment of the $c$ axes with a higher velocity compared to other directions. This COF-derived seismic anisotropy has been investigated in several previous studies (Blankenship and Bentley, 1987; Diez et al., 2015; Kerch et al., 2018). When not





considering this velocity anisotropy effect, a borehole coordinate adjustment scheme could at least partially compensate this
effect.

If a significant velocity anisotropy is present, this could lead to flawed adjustments of the borehole trajectories, which in turn, reduce or entirely remove the anisotropy effect. We have investigated this issue with a synthetic data set. The geometric setup consists of eight straight boreholes (BH01, 02, 04, 05, 07, 08, 10 and 11 in Fig. 1a), and each source hole is again connected to two receiver holes on opposite sides of the ring as in the previous calculations. The true velocity model is set to
a homogeneous velocity value of $3800\,\mathrm{ms}^{-1}$ with an additional ellipsoidal anisotropy of $10\%$ ($\delta = \epsilon = 0.1$) according to the definitions of Thomsen (1986). The azimuthal direction of maximum anisotropy was chosen to be $\vartheta_a = 160°$ with a slightly downwards pointing inclination of $\varphi_a = -15°$. The anisotropy effect was added to the modelled travel times by adding a time shift depending on the angle between maximum anisotropy ($\vartheta_a, \varphi_a$) and the wavefront (derived from the given ray path as defined in Thomsen (1986)). We extended our borehole inversion with additional four model parameters for the anisotropy
parameters and defined a set of estimated values close to but slightly below the true values (i.e. $\delta = \epsilon = 0.05$, $\vartheta_a = 180°$, and $\varphi_a = 0°$). The sensitivities for the four Thomsen parameters were calculated numerically, that is,

$$\frac{\partial t}{\partial m_i} = \frac{f(m_i + \Delta m_i) - f(m_i)}{\Delta m_i}, \tag{10}$$

with initial estimates for the four parameters $m = (\delta, \epsilon, \vartheta_a, \varphi_a)$ and a small perturbation $\Delta m$ for each value. $f(m_i)$ and $f(m + \Delta m)$ are results for the calculated travel times when considering the estimated and perturbed anisotropy parameters,
respectively. When applying this extended inversion algorithm to a data set with the correct values for the isotropic velocity and borehole trajectories, the true anisotropy parameters could be obtained within a few iteration steps. However, when considering straight boreholes as start trajectories, the tests provide evidence that anisotropy parameters are highly dependent from the estimated borehole parameters. The corresponding areas in the resolution matrix for such an experimental setup (Fig. 10) are occupied with large values that indicate a dependency between these model parameters. In addition, Fig. 10 also shows
that especially these model parameters of the borehole trajectories that represent the higher order polynomials significantly interfere with the anisotropy values. Especially the azimuthal angle of anisotropy interferes with the higher order coefficients of all boreholes. This implies that the anisotropy can be erroneously caught by an adjustment of the borehole trajectories. In our synthetic example, this interaction manifests itself in very small adjustments of the anisotropy parameters when the degree of the polynomials is large enough. We observe this effect when considering polynomials of degree $x^4$ or higher. In
contrast, polynomials of degree $x^3$ or lower determine almost correct values ($< 0.01$ for $\delta$ and $\epsilon$ and $< 1.5°$ for the angles) for the anisotropy parameters. However, smaller order polynomials only provide a limited degree of freedom in order to explain deviations along the trajectories. For our synthetic data set, polynomials of degree $x^3$ show optimal results for a simultaneous adjustment of anisotropy and borehole trajectories. We repeated these experiments with different deviations of the borehole trajectories and a range of initial values for the anisotropy parameters. All calculations lead to the same result providing evi-
dence that polynomials of degree $x^3$ provide the best estimates. Higher order polynomials incorporate the anisotropy into the trajectories and the anisotropy parameters are hardly adjusted. In turn, smaller order polynomials suffer from a lower degree of freedom as mentioned above. As a conclusion, these observations confirm that anisotropy and borehole adjustment interfere



with each other and the inversion parameters must be selected in such a way that an anisotropy that might be present in the analysed data set is not accidentally factored out by borehole adjustments.

An application to our field data is difficult to assess since the azimuth and inclination of the COF varies with depth as described in our previous study Hellmann et al. (2021b). Therefore, it was not possible to define a set of Thomsen parameters that describe the seismic velocity anisotropy derived for the COF of the entire ice column. Potentially, crosshole experiments in polar ice with a rather constant COF, which allow for a definition of a set of Thomsen parameters for larger parts of the ice column, are recommended here for further investigations.

## 7 Conclusions


In this paper, we presented a mathematically simple, but efficient approach for a combined velocity and borehole trajectory inversion. This inversion scheme is especially useful for measurements in a deforming subsurface such as experiments in alpine glaciers or ice sheets. The inversion process consists of a typical three-dimensional velocity inversion and an additional borehole trajectory adjustment. All sources and receivers in each borehole are summarised along a trajectory that is characterised

by two orthogonal polynomials. The two steps for velocity and borehole inversion are decoupled as shown by the resolution matrix and thus, we only consider further adjustments of the borehole trajectories if this reduces the entire RMS error of the system. With this sequential inversion, we could significantly reduce velocity artefacts that are a result of poorly determined source and receiver positions.

The adjustments are plausible for our field data but we have also shown that poorly constraint boreholes, e.g. a small amount of

sources and receivers along the borehole, may lead to larger deviations and potentially to an over-fitting of noise and picking errors. Therefore, a data set dependent damping factor which considers a priori information is required. in addition, the algorithm only works for a set of profiles with different azimuths and cannot successfully be applied on two-dimensional profiles. In this case, the polynomial coefficients cannot be determined and oscillations or large deviations occur at the bottom of the boreholes. Nevertheless, we could also show that the coupled scheme is rather robust and converges against very similar so-

lutions for different start coordinates for a reasonably well-constraint inversion problem. We also judge that this approach can be easily applied to other cross-borehole experiments or VSP experiments on solid ground with poorly constrained borehole trajectories. In addition, this approach can in principle be applied to any geophysical borehole experiment beyond seismics. However, its effectiveness needs to be individually studied and validated.

We have also analysed the interdependence between anisotropy and borehole trajectory adjustments. Excessive corrections of

the borehole trajectories can at least partially compensate for the inherent velocity anisotropy. A higher degree of freedom (or flexibility) in the borehole parameters, that is, the use of higher order polynomials, leads to a compensation for any weak ellipsoidal anisotropy. Therefore, it is highly recommended to precisely determine the borehole trajectories during data acquisition. If there is no such a priori information available, the inversion parameters must be selected accordingly to avoid this issue. We have shown that a polynomial degree of $x^3$ provides a good compromise between flexibility to account for changes along the

trajectories and an over-compensation due to an existing velocity anisotropy.



*Code and data availability.* The synthetic and field data are available in the open-access database ETH Research Collection (Hellmann, 2022a). The inversion code is available upon request.

*Video supplement.* The video supplementary material mentioned in the text is available in the open-access database ETH Research Collection (Hellmann, 2022b).

*Author contributions.* This study was initiated and supervised by HM and AB. SH and HM developed the framework. SH and CP wrote the code for the borehole trajectory inversion and anisotropy investigations. SH, MG and AB planned the field campaign and acquired the borehole seismic field data. The data processing was obtained by SH and MG. The paper was written by SH, with comments and suggestions for improvements from all co-authors.

*Competing interests.* The authors declare that they have no conflict of interest.

*Acknowledgements.* This project is funded by the Swiss National Science Foundation (SNSF) under the SNF Grants 200021_169329/1 and 200021_169329/2. We thank Katalin Havas, Johanna Kerch, Dominik Gräff, and Greg Church for their extensive technical and scientific support during data acquisition and processing.





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




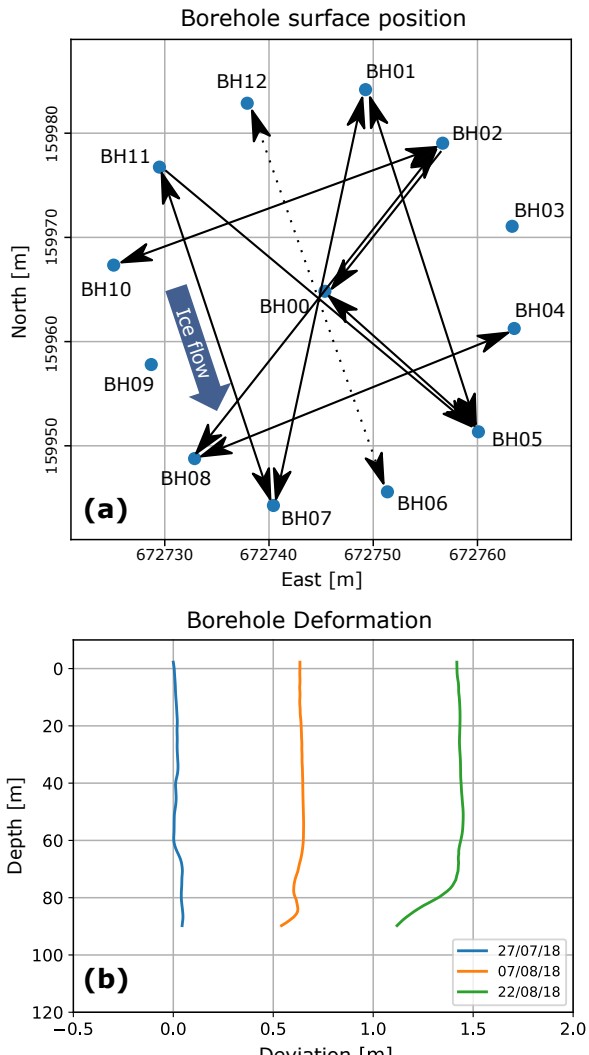

**Figure 1.** Field data measurement geometry. **(a)** Positions of the boreholes on the glacier surface (Swiss coordinates, LV03), BH03, 06, 09 and 12 were not used for the experiments shown here; **(b)** Deformation of borehole BH01 within four weeks due to glacier flow. The profile from 27/07/18 was measured three days after drilling.

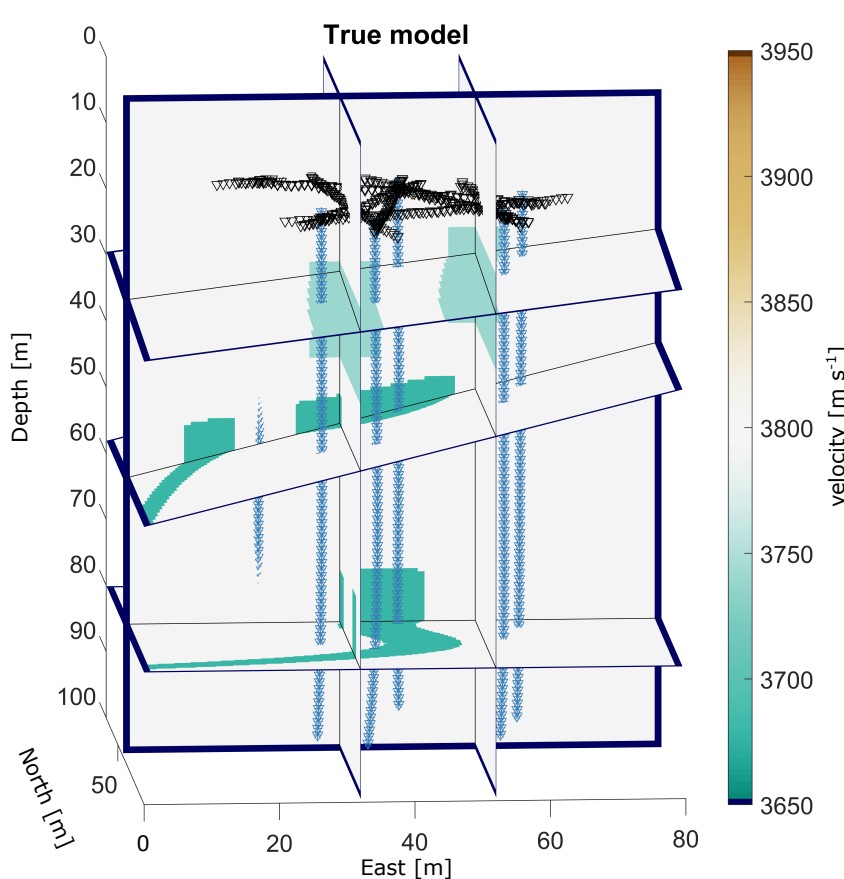

**Figure 2.** True model of the synthetic example data set with a homogeneous background velocity of $3800\,\mathrm{m\,s^{-1}}$ and two vertical fault zones with $v_p$=$3740\,\mathrm{m\,s^{-1}}$ in the upper part and two channels with $v_p$=$3680\,\mathrm{m\,s^{-1}}$ in the deeper parts of the model. Triangles and asterisks represent receiver and source positions in inclined boreholes (blue) and on the surface (black).





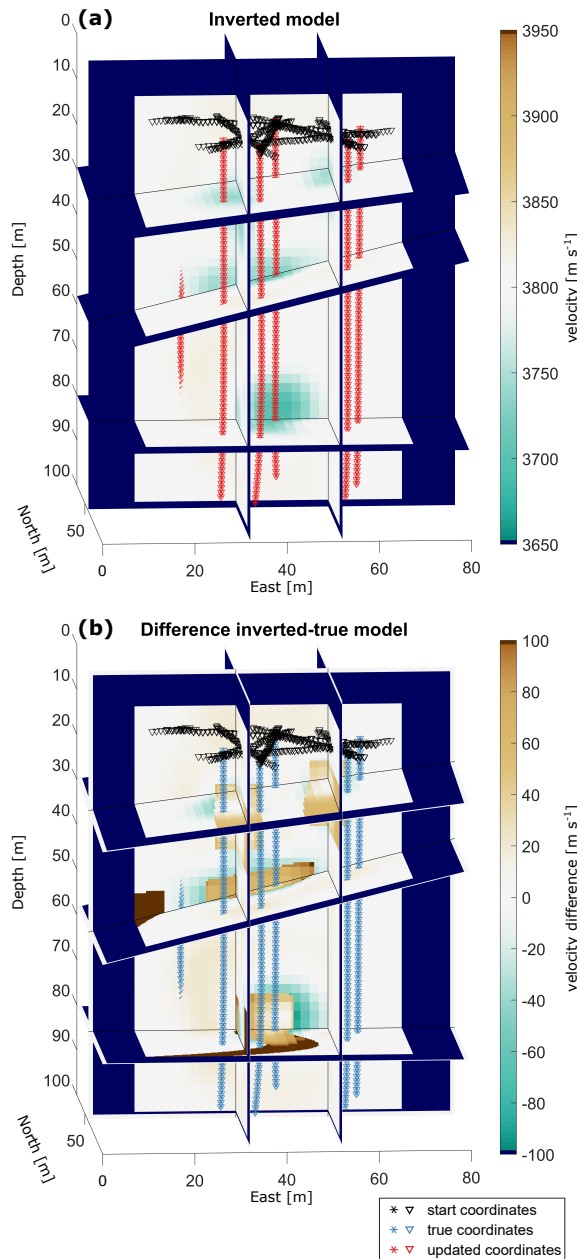

**Figure 3.** Results of the velocity inversion with a synthetic example data set and correct borehole trajectories (a) velocity results of the inversion after 7 steps of iteration; (b) difference between inverted and true model (Fig. 2). Triangles and asterisks represent receiver and source positions.



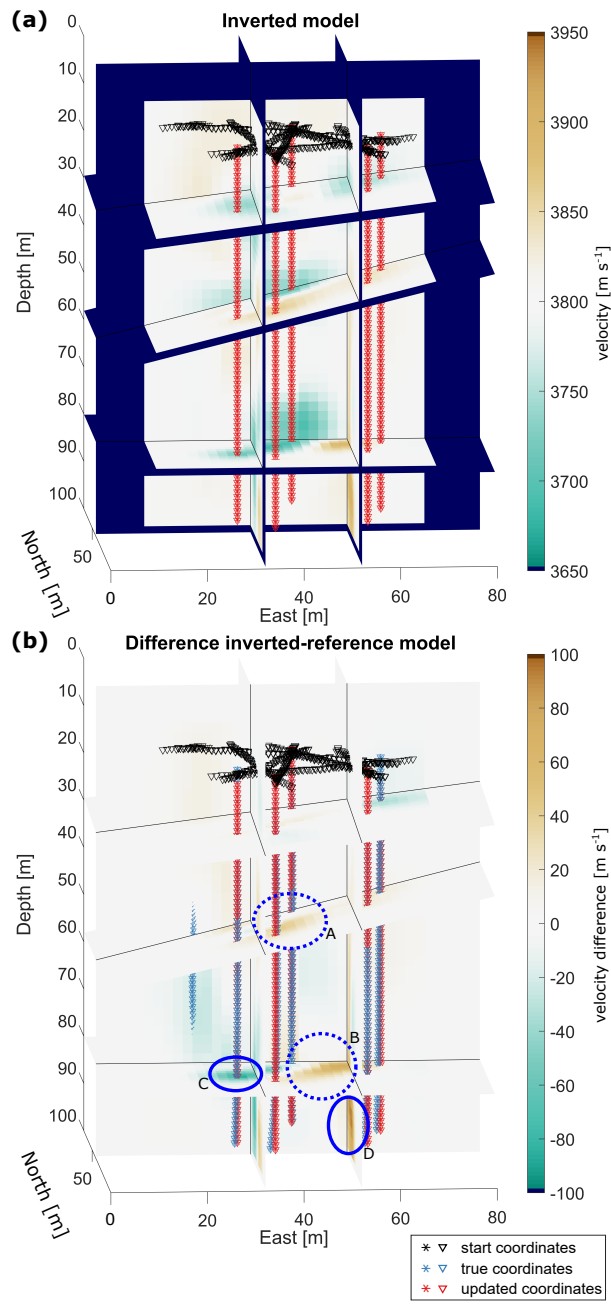

**Figure 4.** Results of the velocity inversion with a synthetic example data set and initially straight borehole trajectories. **(a)** Velocity inversion results after 7 steps of iteration; **(b)** difference between inversion results in **(a)** and reference model (Fig. 3a), turquoise ellipses emphasise the most significant differences. Triangles and asterisks represent receiver and source positions.



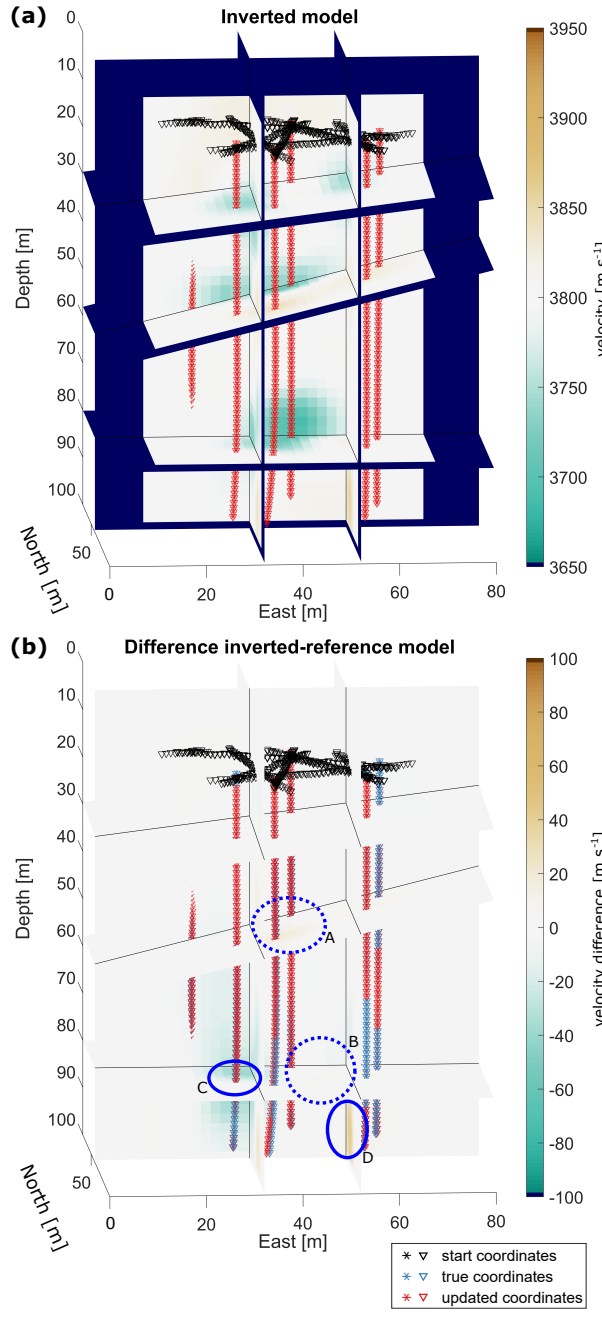

**Figure 5.** Results of the sequential (velocity and borehole trajectory) inversion with a synthetic example data set and initially straight borehole trajectories. **(a)** Inversion results after 7 steps of iteration; **(b)** difference between inversion results in **(a)** and reference model (Fig. 3a). Triangles and asterisks represent receiver and source positions.




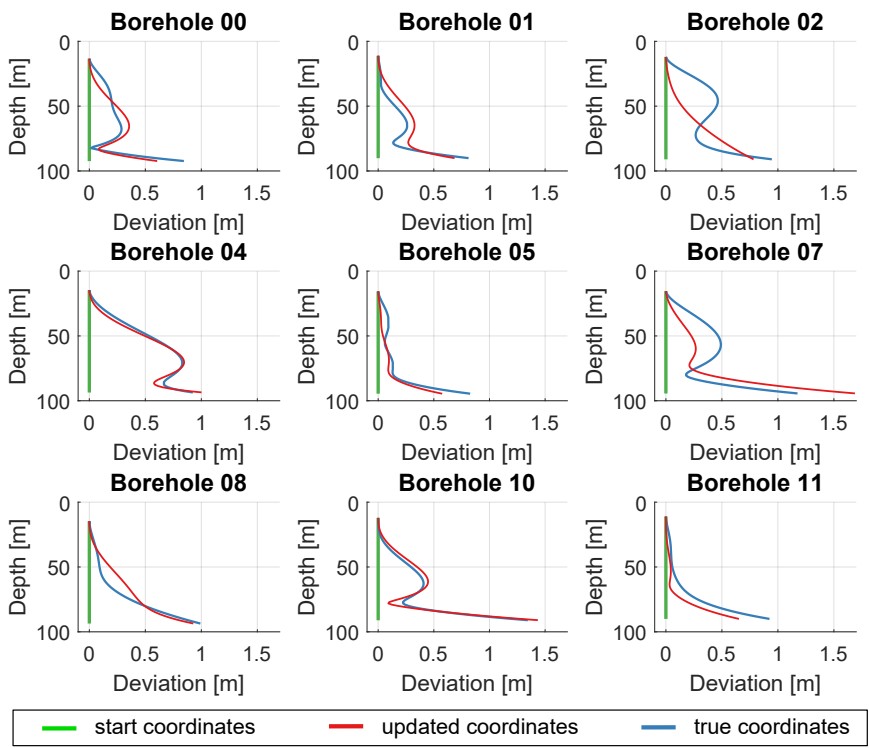

**Figure 6.** Deviations of the individual boreholes (locations shown in Fig. 1a) for the synthetic data set: start values for straight boreholes are plotted as green graphs, true borehole trajectories are shown as blue graphs and the borehole deviations adjusted after seven iterations are plotted as red curves.



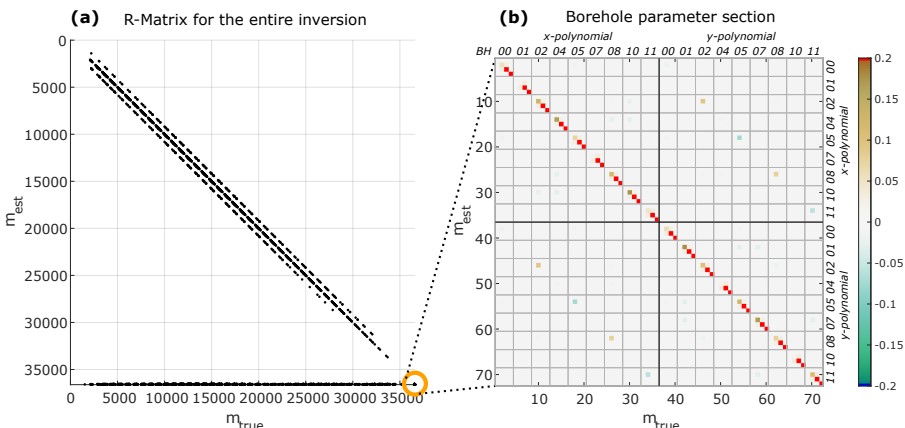

**Figure 7.** Resolution matrix for the coupled velocity and borehole trajectory inversion. Left: the entire matrix for all 36612 model parameters, the 72 parameters of the borehole adjustments are highlighted in the orange circle. Only those values of the matrix are shown as black dots that are larger than 0.05; right: zoomed view to the borehole model parameters with colour-coded values.



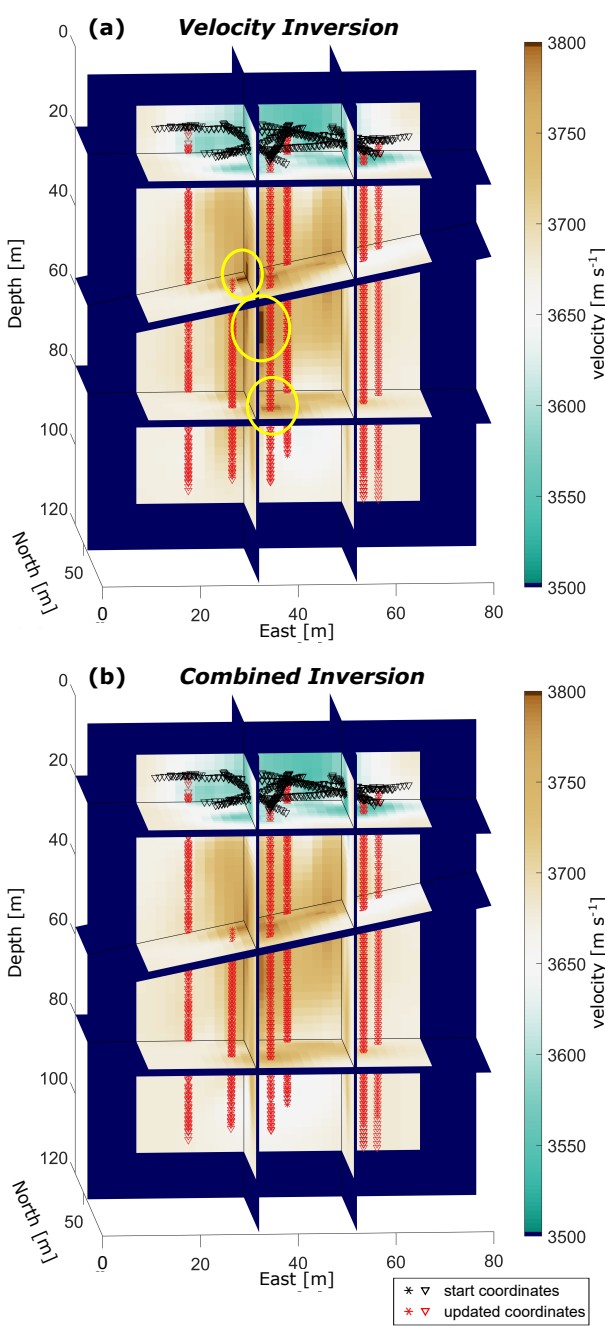

**Figure 8.** Inversion results for field data from Rhonegletscher. **(a)** Results from the velocity inversion (7 iteration) without adjustment of the boreholes (trajectories from inclinometer data); **(b)** results from the combined velocity and borehole inversion with inclinometer-derived start trajectories. Yellow ellipses in **(a)** emphasise the areas with potential velocity artefacts.



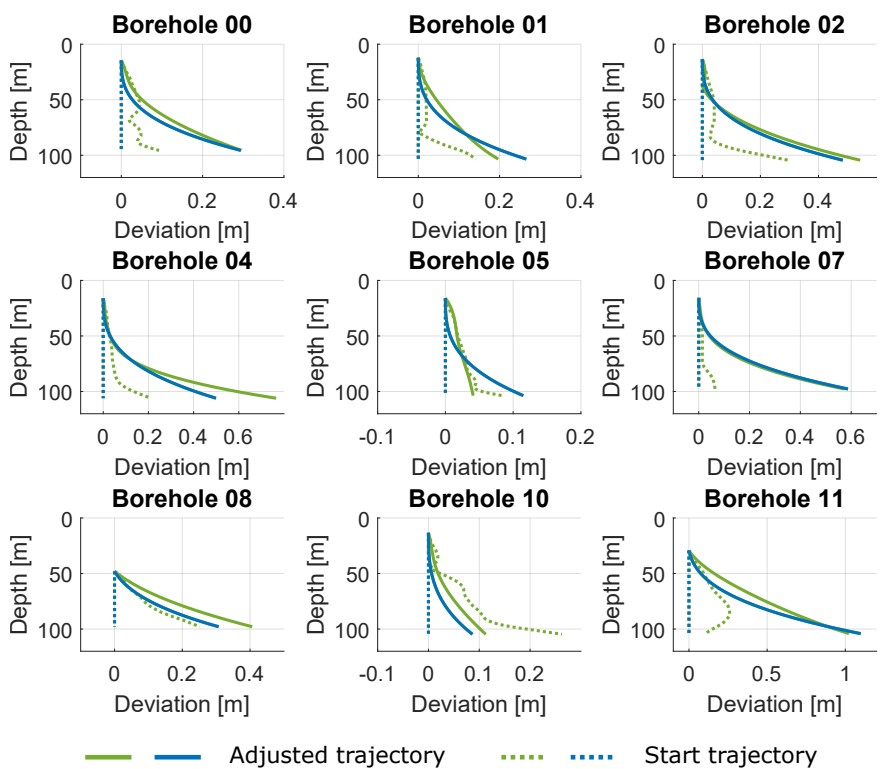

**Figure 9.** Results of borehole trajectory adjustment for field data; green profiles show initial (dashed line) and final (solid line) borehole trajectories for a setup considering a-priori inclinometer data; blue profiles show the borehole trajectories estimated from initially vertical (straight) boreholes.



**Figure 10.** Resolution matrix for a combined inversion for borehole trajectories (third order polynomials) and four anisotropy parameters (marked with TP for Thomsen parameters). The colours show the dependence of the true model parameter from estimated values. Off-diagonal values indicate an interference between different model parameters. The four sectors provide an enhanced overview to distinguish between borehole coefficients and Thomsen parameter (TP).