# Peer review of "A borehole trajectory inversion scheme to adjust the measurement geometry for 3D travel time tomography on glaciers"

_EGUsphere, 2022_

## Author Comment (AC1)

Dear anonymous reviewer,

Thank you very much for the valuable comments. We will consider them in our final version. Please find our detailed answers to your comments below.

- Section 3
    - It is better to show the day of data acquisition because its timing is discussed in Section 5. It would be easier for the reader to understand and evaluate the results in this manuscript.

We assume your comment refers to the legend of Fig. 1? We have considered this suggestion in the final version of this figure.

- Section 3, line 169
    - It is not clear what the author trying to describe by "Geophones at the surface of the glacier complemented the measurement setup.". It would be better to mention the purpose of geophone and their usage in this study (included in the inversion?).

Agreed. We have added additional details to make it more obvious that these geophone lines between the source and the two receiver boreholes are also included in the inversion for an enhanced ray coverage:
"Geophones, equally spaced at 1m between the source and receiver boreholes, were placed at the surface of the glacier. These additional receivers further increase the azimuthal resolution of the tomographic experiment.  The enhanced ray coverage in the area of investigation reduces the ambiguities between model parameters."

- Section 4, lines 252 – 254
    - It is difficult to understand which part of Fig. 7b is mentioned in this sentence. It would be better if the author describes the more detail of the figure and explain which part is polynomial coefficients one through 4th degree. Also, it is difficult to see which parts are a value > 0.99 since the color bar is monotone above the value > 0.2.

We have rephrased the respective sentences, and added additionally: "For the borehole inversion parameters, the relationship between estimated and true parameters is much stronger, especially for the polynomial coefficients of degree one and two. As shown in Fig. 7b, the third and fourth element of each set of borehole coefficients representing degree one and two, show high values > 0.99. Therefore, these coefficients are well resolved in the inversion."
In conjunction with a comment of the other reviewer, the colour bar has been refined.

- Sections 4 and 5
    - Although the number of polynomials in the inversion is discussed in 6.2, the author does not show the value in their application to synthetic and field data. Its value is important for anisotropic effect according to 6.2. Please include the number of polynomials that are used for inversions in the manuscript.

We have added the respective numbers for the synthetic data ("Each borehole is approximated by a set of two mutually perpendicular polynomials of degree four.") and an additional table showing the number of coefficients for both polynomials of each borehole used for the field data (polynomials of degree 3-5).

- Section 5, line 304
  - This sentence explains that the author used two starting models for trajectory inversion. However, the result of interpolated trajectory case (second starting model) is already described in Fig.8b in line 289. It is easier for a reader to understand if the prerequisite is described before or just after its results are shown.

In our revised version, we have rearranged the respective lines and provide the information, where considered the first time.

- Section 5, line 305
  - According to the caption of Figure 8b, it is the result of using interpolated trajectories as an initial model. However, this sentence says that Figure 8b is the result using vertical boreholes as a starting model. It is inconsistent. The results of using vertical wells as an initial model are not shown in this manuscript. Please add the results.

We add a Fig 8c showing the results of the inversion with initially vertical boreholes. In this context, we have revised the caption of Fig. 8 and add the correct references in line 305.

- Section 6.1 line338
  - "lower that" should be "lower than".

Thank you, we corrected this typo.

- Figures 3, 4, 8 (a)
  - Legends in this figure are confusing since these are velocity inversion results without coordinates updated. The expression "updated coordinates" seems not right.

Indeed, the wording is inconvenient here. We have considered this point during the revisions.

- Figures 3, 4, 5
  - It seems black asterisks and triangles in Figures 3 to 5 are coordinates for geophones. Please describe what "start coordinates" in the legends mean. The term "start coordinates" is also used in Figure 6, but it seems different from figures 3 to 5.

We have considered these points in our reviewed version.

Kind regards,

Sebastian Hellmann and Hansruedi Maurer
(on behalf of the co-authors)

---

## Author Response (AR1)

*Point-to-point response to the reviewers comments:*

**Reviewer comment, 02 Dec 2022:**

- Section 3
    - It is better to show the day of data acquisition because its timing is discussed in Section 5. It would be easier for the reader to understand and evaluate the results in this manuscript.

We assume your comment refers to the legend of Fig. 1? We have considered this suggestion in the final version of this figure.

Changes in Fig. 1:
We have changed the legend of Fig. 1 using the concrete days after drilling rather than dates.

- Section 3, line 169
    - It is not clear what the author trying to describe by "Geophones at the surface of the glacier complemented the measurement setup.". It would be better to mention the purpose of geophone and their usage in this study (included in the inversion?).

Agreed. We have added additional details to make it more obvious that these geophone lines between the source and the two receiver boreholes are also included in the inversion for an enhanced ray coverage:

Changes to the text (lines 184ff):
"Geophones, equally spaced at 1m between the source and receiver boreholes, were placed at the surface of the glacier. These additional receivers further increase the azimuthal resolution of the tomographic experiment. The enhanced ray coverage in the area of investigation reduces the ambiguities between model parameters."

- Section 4, lines 252 – 254
    - It is difficult to understand which part of Fig. 7b is mentioned in this sentence. It would be better if the author describes the more detail of the figure and explain which part is polynomial coefficients one through 4th degree. Also, it is difficult to see which parts are a value > 0.99 since the color bar is monotone above the value > 0.2.

We have rephrased the respective sentences, and added additionally:

Changes to the text (lines 281ff):
"For the borehole inversion parameters, the relationship between estimated and true parameters is much stronger, especially for the polynomial coefficients of degree one and two. As shown in Fig. 7b, the third and fourth element of each set of borehole coefficients representing degree one and two, show high values > 0.99. Therefore, these coefficients are well resolved in the inversion."

Changes in Fig. 7:
The colour bar has been refined in such a way that values close to 1 are now coloured in violet. In addition, a zoomed excerption has been added to show the order of the polynomials for a borehole as an example.

- Sections 4 and 5
    - Although the number of polynomials in the inversion is discussed in 6.2, the author does not show the value in their application to synthetic and field data. Its value is important for anisotropic effect according to 6.2. Please include the number of polynomials that are used for inversions in the manuscript.

We have added the respective numbers for the synthetic data and an additional table showing the number of coefficients for both polynomials of each borehole used for the field data (polynomials of degree 3-5).

Changes to the text (line 250f):
"Each borehole is approximated by a set of two mutually perpendicular polynomials of degree four."

Changes: new Table 3

- Section 5, line 304
    - This sentence explains that the author used two starting models for trajectory inversion. However, the result of interpolated trajectory case (second starting model) is already described in Fig.8b in line 289. It is easier for a reader to understand if the prerequisite is described before or just after its results are shown.

In our revised version, we have rearranged the respective lines and provide the information, where considered the first time.

Changes to the text (line 319ff):
"We considered two slightly different start models. The first one uses the trajectories derived from inclinometer measurements as start values. The results are shown in Fig. 8b. The second start model considers vertical boreholes. The differences between both are discussed below and are rather small. Therefore, we did not include the results here but provide a difference plot in the appendix (Fig. A1). The velocity and the two combined inversion models show a low-velocity zone close to the surface, which we interpret as a zone of weathered, water-saturated ice."

- Section 5, line 305
    - According to the caption of Figure 8b, it is the result of using interpolated trajectories as an initial model. However, this sentence says that Figure 8b is the result using vertical boreholes as a starting model. It is inconsistent. The results of using vertical wells as an initial model are not shown in this manuscript. Please add the results.

We add a Figure in the appendix showing the differences between both start models.

Changes: new Fig. A1 in the appendix

Changes to the text (lines 334f):
"Figure 8b shows the results for an inversion with borehole trajectories interpolated from inclinometer measurements. We also repeated the calculations for initially vertical borehole trajectories. The results are nearly identical within a range of 40 m/s as shown in the difference plot in Fig. A1."

- Section 6.1 line338
  - "lower that" should be "lower than".

Thank you, we corrected this typo.

Changes to the text: replacing t by n

- Figures 3, 4, 8 (a)
  - Legends in this figure are confusing since these are velocity inversion results without coordinates updated. The expression "updated coordinates" seems not right.

Indeed, the wording is inconvenient here. We have considered this point during the revisions.

Changes to the Figures: Legends have been revised and adjusted as recommended.

- Figures 3, 4, 5
  - It seems black asterisks and triangles in Figures 3 to 5 are coordinates for geophones. Please describe what "start coordinates" in the legends mean. The term "start coordinates" is also used in Figure 6, but it seems different from figures 3 to 5.

We have considered these points in our reviewed version.

Changes to the Figures: We removed the misleading term "start coordinates" and the asterisks and renamed the triangles to "surface geophones".

**Reviewer comment, 17 Feb 2023:**

Abstract:

- In the abstract it sounds like no a priori information are necessary anymore, but as stated in the conclusion, selection of damping parameter and polynomial order is still based on deviation measurements.

We have rephrased the respective part of the abstract.

Changes to the text (line 15ff):
"In combination with a rough approximation of the borehole trajectories, for example, from additional a priori information, heterogeneities in the velocity model can be imaged similar to an inversion with fully correct borehole coordinates."

- Line 20: I suggest to replace "Therefore" with "Based on the modelling results we propose/determined", because this is not the conclusion of the previous explanation, but of the investigation you described in the manuscript.

Agreed.

Changes to the text (line 21):
"Based on the modelling results, we propose to consider polynomials up to degree three."

- Line 22: Compensation for velocity anisotropy is minimized not avoided.

Agreed.

Changes to the text (line 23):
"... and minimising compensation for velocity anisotropy."

Section 3:

- It is not clear when the seismic acquisition happened and if the deviation measurements were done on the same days. An overview in form of e.g. a small table would help. As I understood there was ~ 10 days between the acquisition days. It does not get clear how the inversion incorporates borehole deviation for the same borehole on different days. As I understand the polynomial coefficients are adjusted with the whole dataset, but the deviation is different for the subsets collected on different acquisition days. Please give insight on how this is done.

In principle, a separate borehole trajectory would have to be computed for the acquisition of every tomographic plane, because there are ongoing movements and deformations of the glacier. However, this would lead to a poorly constrained inversion problem. Instead, we have designed our experimental schedule, such that a single borehole is occupied for a minimal time span (up to 4 days), and we assume that the trajectory changes within this short time span are acceptably small. We acknowledge that this may lead to minor inaccuracies, but, in our view, they are unavoidable. We have clarified this in the revised text.

Changes to the text (lines 169ff):
The continuous displacements and deformations of the glacier ice body are still imposing problems in our attempt to invert simultaneously for subsurface velocities and the borehole trajectories. In principle, the borehole trajectories would have to be estimated for every single source-receiver borehole pair, but this would lead to a poorly constrained inversion problem. To avoid this problem, we have set up our experimental schedule, such that every borehole is only occupied for a relatively

short time span (up to four days). We then made the assumption that the changes of the trajectories are acceptably small within this short time span (i.e., we have inverted for a single trajectory for each borehole). This is certainly a limitation of our methodology, but it is, in our view, an unavoidable compromise that needs to be made."

- The model includes anomalies that are not resolved by the tomography, because there is no ray coverage in this area. When an anomaly like this is incorporated in the model this should be mentioned in the discussion of the results to avoid confusion.

Indeed, some of the anomalies remain unresolved, because they are not covered adequately with rays. We have added a clarifying comment.

Changes to the text (lines 231ff):
"In turn, this implies that the artefacts in those parts of the model lying outside the ring of boreholes and close to the bottom could therefore not correctly be inverted by none of the inversions and thus will be excluded from the following discussion."

- Line 169: How many geophones are installed at the surface? In which setup? It is also not clear whether the data from the surface geophones is used in the inversion. This would also make the model setup in line 187 more understandable ("…and added a total of 828 receivers…")

We describe the setup with some additional details in the revised text.

Changes to the text:
lines 184ff: "Geophones, equally spaced at 1 m between the source and receiver boreholes, were placed at the surface of the glacier. These additional receivers further increase the azimuthal resolution of the tomographic experiment. The enhanced ray coverage in the area of investigation reduces the ambiguities between model parameters."

lines 205ff: "These geophones where installed at the glacier surface along 2D lines between the source and receiver boreholes. During data acquisition, about 95 geophones were active at a time, i.e. those along the lines between the currently used boreholes."

- Line 172: "…explained in more detail below." State where it is explained "…explained in more detail in section 4."

Added.

Changes to the text (line 189):
"... as explained in more detail in Sect. 4."

Section 4:

- Line 201: The damping factor of 0.1 is selected. In Line 300 a damping factor between 1000 and 10000 is recommended. It is not clear if these are different factors and why the magnitude changes.

Indeed, our formulation in the original text is rather confusing. The damping factors indicated are relative numbers with relatively little physical meaning (they relate to the magnitudes of the sensitivities of the individual model parameters). After considering a variety of options, we finally concluded that it would be most useful to rather describe our regularisation strategy instead of providing numbers that are difficult to interpret.

Changes to the text (lines 219ff):
"Since the tomographic inversion problem includes a significantly underdetermined component, it is necessary to provide regularisation constraints. For the velocity parameters, they were supplied in form of damping and smoothing constraints (Maurer et al., 1998). After some experimentation, we applied similar weights for damping and smoothing (damping factor of 0.1, smoothing factor of 0.4). We tuned the overall regularisation factor, such that the inversions converged to a data misfit level that corresponds roughly to the travel time picking accuracy. For the model parameters associated with the borehole trajectories, we employed only damping constraints, with which we penalised deviations from the initial values. We started with relatively high damping factors, which we gradually reduced, until the inversions became unstable, and/or unreasonably large deviations were obtained."

- Table 1: For evaluation of the combined inversion adding the RMSE of velocity inversion with true trajectory should be a benefit.

We have considered this during our revisions.

Changes to Table 2:
New line showing the RMS errors for the reference model.

- Line 220: Here add a reference to the supplementary video material [Hellmann,2022a]

Added.

Changes to the text (line 260):
Reference added.

- Line 215 and 224: I do not agree with the statements, that in Figure 4 the lower channel "could not be recovered", while in the combined inversion in Fig. 5 "the upper and especially the lower channel are correctly resolved". The anomaly associated with the channel is well visible in both results. I agree, that there is less artefacts and the channel is better resolved in the combined inversion results which supports the papers objective. Please clarify this.

When comparing the differences between true and inverted data, they are very large for the lower channel without borehole adjustments. Nevertheless, we agree that the velocity inversion also roughly recognises the channel. Therefore, we mitigate the statement "could not be recovered" to e.g. "is only weakly recovered".

Changes to the text (line 215):
"is only weakly recovered"

- Line 247 / Fig. 7b: It is not clear which values are in the range of >0.99 since the colorbar ends at 0.2. The allocation of the polynomial coefficients on x- and y-axis is not intuitive. Label the polynomial coefficients on the axes. For better visibility only a fragment of the matrix could be shown in order to gain the space for labels.

We need to evaluate the options how to improve this figure. An additional excerpt showing the values for one borehole could be an option. However, just showing a part of Fig. 7b seems to be not a valid option. From our point of view, we need to show that the mentioned coefficients of lower degrees along the main diagonal are well resolved for all boreholes as discussed in the text.

Changes to Fig. 7b:
Colour scale has been adjusted and an excerpt has been added showing the order of the polynomial coefficients as an example for one borehole polynomial.

Section 5:

- 279: Is 200ms correct? A 200ms window around the estimated arrival time seems very long since the highest expected traveltimes at v=3800m/s and max. distance of 100m (90m depth, 40m borehole spacing) are ~25ms.

Thank you very much for finding this mistake. Indeed, this is a wrong value and must be 4 ms. We used a window of min. 40 to max. 200 *samples* (depending on the distance between source and receiver) and the sampling rate was 20.833 us.

Changes to the text (lines 308f):
"[For the field data set, we determined the travel times of the recorded p-waves with a cross-correlation algorithm between repetitive measurements (in general 3 repetitions) within a] window with a length of up to 4 ms depending on the distance around the estimated arrival time."

- Line 312: Deviations of 0.6m and 1m are seen as not realistic, while having a displacement speed of 0.06m/d and 11-14 days. This results in 0.06m/d*14d = 0.84m which seems realistic in this context.

No, this is not correct. The ice flow rate in valley glaciers usually decreases with depth. This is common sense in the glaciology community, but we should clarify this with an additional clause:
Changes to the text (line 349):

"[Glacier flow rates of 0.06 m d$^{-1}$ on average measured at the glacier surface during the summer period imply that this deviation could not be caused by the ice flow], when considering a typical parabolic decrease of the flow rate with depth."

Section 3 and 5:

- What is the estimation of the borehole deviation measurement error? How does it compare to the difference between measured borehole trajectory and fitted trajectory after inversion? The comparison is important to assess the plausibility of the trajectory fitting.

This is a very difficult question. We tried to determine the borehole trajectories with various tools, and we observed discrepancies, well beyond the accuracy, specified by the manufacturers. This was the main motivation to perform this research, and to develop the inversion strategy. However, the large discrepancies observed, make it very difficult to specify the accuracy of the initial values.
We have added a corresponding statement in the introduction.

Changes to the text (already in lines 91ff):
"At the same time, we found that the precision of different in struments was not sufficient under the given measurement conditions on glaciers. In some cases, significant deviations occurred between the different instruments, although measurements were carried out directly one after the other. These observations initially triggered our investigations for a joint velocity and coordinate inversion."

Section 6:

- Line 328: "However, there is a risk that the coordinate adjustment will suppress the appearance of real velocity anomalies in the tomogram. We avoid this by decoupling the two parts of the inverse algorithm." Is it avoided? At the end of Section 2 Line 140 you explain that sequenced inversion and the inversion with extended set of equations do not show significant differences is the results. Could you please explain this.

The advantage of the sequential inversion scheme lies in its flexibility as described in the article. For the synthetic tests, we compared the sequential scheme with a common inversion scheme that inverts for all model parameters (i.e. velocity and polynomial coefficients) simultaneously. In the comparison, we always updated the coordinates in both schemes. However, when later applying the sequential scheme, other constraints can be considered whether an update in the current step of iteration is useful (i.e. further reduces the RMSE). If so, the new coordinates derived from the updated polynomials are used. This provides the additional and mentioned flexibility compared to a common inversion that updates all model parameter thus providing less flexibility.
We have added another sentence in Lines 150f to make this more clear.

Changes to the text (already in line 150/151):
"It also provides more flexibility to decide if an update of the coordinates in the current step of iteration is beneficial and thus applied or skipped."

- Section 6.2: This section gives important insight on trade off between trajectory optimization and anisotropy. A lot of inversion results are presented without visualizing any of them. An additional Figure is recommended. If to many Figures are already in the manuscript than add significant Figures to the supplement.

We add another Figure that demonstrates the influence or interaction between borehole trajectory adjustments and anisotropy.

Changes: adding Figure 10 (already existing Fig. 10 is Fig. 11 now)

Changes to the text:
lines 414ff: "In a first run, we used our combined velocity and borehole inversion scheme to invert for such an anisotropic model starting with vertical boreholes. The resulting borehole trajectories, shown in Fig.10a, incorporate the anisotropy effect up to a certain degree and drift away from the true trajectories."
lines 436f: "Figure 10b shows the results of such an anisotropic inversion with a good fit between estimated and true borehole trajectories."

Section 7:

- Line 439: Video Supplement is in [Hellmann, 2022a], not [Hellmann,2022b]

Thank you, we have exchanged the entries in the References so that line 439 and also line 437 are pointing to the correct reference.

Changes to the text:
adjusting the bibtex-key for the two references.

- If possible give an outlook on how to determine inversion parameters (damping, …) without a priori information from inclinometer measurements. If these are always required the advantages of the combined inversion are mitigated.

We have added a section in the conclusion. We add two possible ways how to better determine the regularisation parameters.

Changes to the text:
"A weakness of our inversion methodology includes the somewhat subjective choice of the regularisation parameters. In future investigations, this may be improved in two ways. When more advanced information on the glacier movements would be available, this could be supplied in form of further constraints to the inversion problem. Alternatively, the various regularisation parameters could be determined in a more systematic fashion (compared with our trial-and-error approach). A possible option may include generalised cross-validation (e.g. Golub et al., 1979) or similar techniques."